# Phenotypic states become increasingly sensitive to perturbations near a bifurcation in a synthetic gene network

Kevin Axelrod[1], Alvaro Sanchez[2], Jeff Gore[3]*

[1]Harvard University, Graduate Program in Biophysics, Cambridge, United States; [2]Rowland Institute, Harvard University, Cambridge, United States; [3]Department of Physics, Massachusetts Institute of Technology, Cambridge, United States

**Abstract** Microorganisms often exhibit a history-dependent phenotypic response after exposure to a stimulus which can be imperative for proper function. However, cells frequently experience unexpected environmental perturbations that might induce phenotypic switching. How cells maintain phenotypic states in the face of environmental fluctuations remains an open question. Here, we use environmental perturbations to characterize the resilience of phenotypic states in a synthetic gene network near a critical transition. We find that far from the critical transition an environmental perturbation may induce little to no phenotypic switching, whereas close to the critical transition the same perturbation can cause many cells to switch phenotypic states. This loss of resilience was observed for perturbations that interact directly with the gene circuit as well as for a variety of generic perturbations-such as salt, ethanol, or temperature shocks-that alter the state of the cell more broadly. We obtain qualitatively similar findings in natural gene circuits, such as the yeast GAL network. Our findings illustrate how phenotypic memory can become destabilized by environmental variability near a critical transition.

*For correspondence: gore@mit.edu

Competing interests: The authors declare that no competing interests exist.

## Introduction

Microbes such as yeast and bacteria often adopt a specific phenotype in response to an environmental stimulus, and in many cases this phenotype can be retained even after the stimulus has been removed (*Ozbudak et al., 2004*; *Acar et al., 2005*; *Hu et al., 2012*; *Williams et al., 2013*). Stably storing such a phenotype can be critical for survival during adverse environmental changes (*Gasch et al., 2000*; *Gasch and Werner-Washburne, 2002*; *Chen et al., 2003*; *Brauer et al., 2008*). Several well-studied natural gene networks can be found in multiple phenotypic states depending on the history that the cells have experienced, notably including the yeast galactose utilization network and sporulation commitment in *Bacillus subtilis* (*Acar et al., 2005*; *Igoshin et al., 2006*). Additionally, synthetic biology aims to create de novo phenotypic memory storage devices, which could be used in applications such as tracking dynamics of the gut microbiome (*Ajo-Franklin et al., 2007*; *Kotula et al., 2014*). In spite of the importance of phenotypic memory in natural and synthetic gene circuits, little is known about how environmental perturbations might disrupt a cell's ability to stably adopt a specific phenotype.

Given that phenotypic memory often results from feedback loops within the cell (*Ferrell and Machleder, 1998*; *Pomerening et al., 2003*; *Xiong and Ferrell, 2003*; *Angeli et al., 2004*; *Ozbudak et al., 2004*), it is possible that this phenotypic memory will display characteristics of other complex systems that exhibit bistability, memory, and associated critical transitions or 'tipping points' that lead to sudden changes in the state of the system in response to small changes in the environment. A phenomenon that may be especially relevant to cellular memory is that complex systems near

**eLife digest** All organisms need to be able to react to the challenges thrown at them by their changing environment. Yeast, bacteria and other microbes have networks of genes that can give rise to many different traits and characteristics, which can also be referred to as phenotypes. A change in the environment can alter the activities' of the genes so that the microbes display a different phenotype. The point at which a small change in the environment can lead to a sudden switch in the phenotype is called a 'critical transition'.

An individual microbe's history can influence the phenotype that it presents. However, it is not clear how microbes 'remember' their history, or how fluctuations in the environment might cause the microbe to lose the ability to store this memory and present a different phenotype instead. Here, Axelrod et al. studied phenotype memory in yeast cells grown in the laboratory. The experiments used cells that had been genetically modified to glow red in the presence of a molecule called anhydrotetracycline (or ATc) and to glow green in the absence of the molecule.

Axelrod et al. examined what effect altering the levels of this molecule would have on the phenotype produced by the cells. First, the cells were grown with no ATc present for several generations so that the cells glowed green. Next, Axelrod et al. added different amounts of ATc were added. For moderate levels of ATc the cells continued to glow green, illustrating that they 'remembered' their prior growth condition. However, cells exposed to higher levels of ATc lost this memory and changed color.

Next, Axelrod et al. carried out further experiments on cells exposed to ATc levels that were close to, or further away from the critical transition. At high levels of ATc (that is, close to the critical transition), many cells switched from green to red when exposed to high temperatures, salt and other changes in the environment. On the other hand, very few of the cells grown in low levels of ATc—and therefore further away from the critical transition—changed color in response to the same fluctuations in their environment.

These finding reveal that phenotype memory is less stable when yeast experience fluctuations in their environment close to a critical transition. Future work will seek to find out how salt or high temperatures can abolish phenotype memory.

a critical transition experience a loss of resilience of their stable states to external perturbations (*Dai et al., 2012*). In particular, a system far from a critical transition may return to its original state following a perturbation, whereas closer to the critical transition the same environmental perturbation may cause the system to switch to an alternative stable state (e.g., collapse of a population [*Dai et al., 2012*]). The dynamics in the vicinity of the critical point are very slow on both sides of the critical point. Therefore, it is worth noting that a short lived perturbation that pushes the system past the critical point may also not cause switching if the perturbation is short enough, if it pushes it to a position that is close to the critical point, or both. This aforementioned loss of resilience near a critical transition results from a shrinking basin of attraction in the stability landscape (*Scheffer et al., 2009*). In the context of a gene network, this loss of resilience would manifest as an increasing sensitivity of phenotypic memory against environmental perturbations approaching the environmental condition in which cells would (deterministically) switch to a different phenotype. Our initial goal was to observe memory in a model gene circuit and then characterize the resilience of the phenotypic state against perturbations in the extracellular environment.

## Results

To study this predicted loss of resilience, we first employed a synthetic genetic switch in budding yeast that has previously been shown to exhibit hysteresis and bistability (*Gardner et al., 2000*; *Blake et al., 2006*; *Ellis et al., 2009*; *Wu et al., 2013*). This toggle switch is composed of two mutually inhibitory transcription factors, LacI and TetR (*Figure 1A*). To enable tracking of the state of the cell, different color fluorescent proteins are expressed depending upon which of these transcriptional factors is highly expressed (mCherry and eGFP, hereafter referred to as 'RFP' and 'GFP'). As external knobs to control the state of the cell, the inducer IPTG modulates the strength of repression of LacI, whereas ATc modulates the strength of repression of TetR.

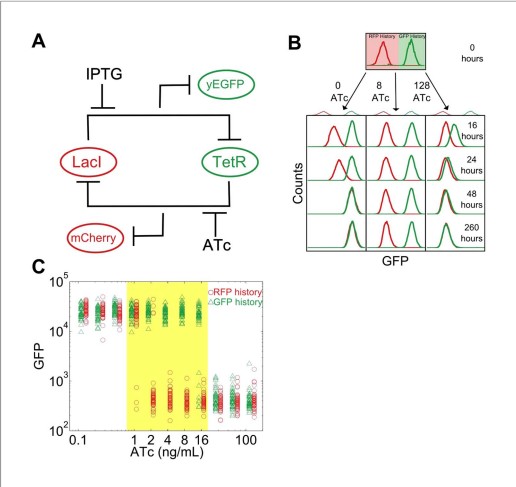

**Figure 1**. A toggle switch in yeast exhibits hysteresis and bistability. (**A**) A toggle switch consists of two mutually inhibitory transcription factors, two fluorescent readouts of the system state, and two small molecule inhibitors of the transcription factors. (**B**) Following growth in one of two histories, cells are then diluted into a range of ATc concentrations and propagated in culture for several days. Histogram counts are binned logarithmically. (**C**) The intensity of GFP fluorescence is plotted as a function of (ATc) for 11 different conditions for the high GFP history (green triangles) and high RFP history (red circles) after 92 hr of growth. The distributions are offset for ease of viewing. 20,000 events are collected, and then a narrow gate is drawn to select several hundred cells of roughly equal size. From this narrow gate, 50 cells at random are plotted. The region of memory is shaded in yellow.

The following figure supplements are available for figure 1:

**Figure supplement 1**. Fraction switched after 92 hr of growth, a proxy for the instability of the state, increases approaching a phenotypic switch.

**Figure supplement 2**. The switching kinetics are non-exponential.

As a demonstration of how cellular memory operates in this gene network, one population of cells was pre-grown in IPTG to initialize the cells in a high GFP state (*Figure 1B*). A separate population of cells was pre-grown in ATc to initialize the cells in a high RFP state. We then transferred the populations to a range of ATc concentrations and monitored the dynamics over several days using flow cytometry. All cultures were grown with a fixed concentration of IPTG (40 µM), which served as an orthogonal control variable for later experiments. For high and low concentrations of ATc, cells converged to a state that was independent of the history they were pre-grown in (*Figure 1B,C*). However, for intermediate concentrations of ATc there was little to no switching of the phenotypic state of the cells even after 260 hr of growth and over 80 cell divisions. The mutual repression in this toggle switch therefore indeed allows for remarkably stable cellular memory for intermediate ATc concentrations. As expected (*Acar et al., 2005*), the lifetime of the states decreases approaching the critical transition at which switching is deterministic (hereafter, we refer to this point as the 'critical transition' or 'phenotypic switch'), although interestingly we observe non-exponential switching kinetics, potentially indicating the presence of metastable states in the gene network (*Figure 1—figure supplements 1, 2*). This decrease in lifetime is one manifestation of deteriorating cellular memory approaching the critical transition, but it is not obvious whether in ATc concentrations with long lifetimes (i.e., strong memory) the cells are also able to retain their memory in the face of environmental perturbations.

In analogy to other complex systems, we hypothesized that phenotypic states in this gene network might become increasingly sensitive to perturbations near a critical transition (*Van Nes and Scheffer, 2007*). This expectation arises because the state's basin of attraction shrinks as the stable and unstable fixed points approach one another (*Figure 2A*). Brief environmental perturbations will push the system out of equilibrium. If the system is far from the critical transition it will return to its original state after the perturbation is removed (*Figure 2B*). However, close to the critical transition, the same perturbation might cause the system to cross the basin boundary, thus causing the cell to switch phenotypic states. Therefore, phenotypic states are expected to lose resilience to environmental perturbations near a critical transition.

To test this theoretically proposed loss of resilience we perturbed cells at different distances from the critical transition (i.e., different ATc concentrations) and measured how likely the cells were to switch into the alternative phenotypic state as a result of environmental perturbations. We examined two classes of perturbations, 'directional' (in which the perturbation increased the probability of switching into the alternative state by inhibiting one or another of the two transcription factors that form the toggle switch) and 'generic' (in which the interaction between the toggle switch and the perturbation was not immediately obvious).

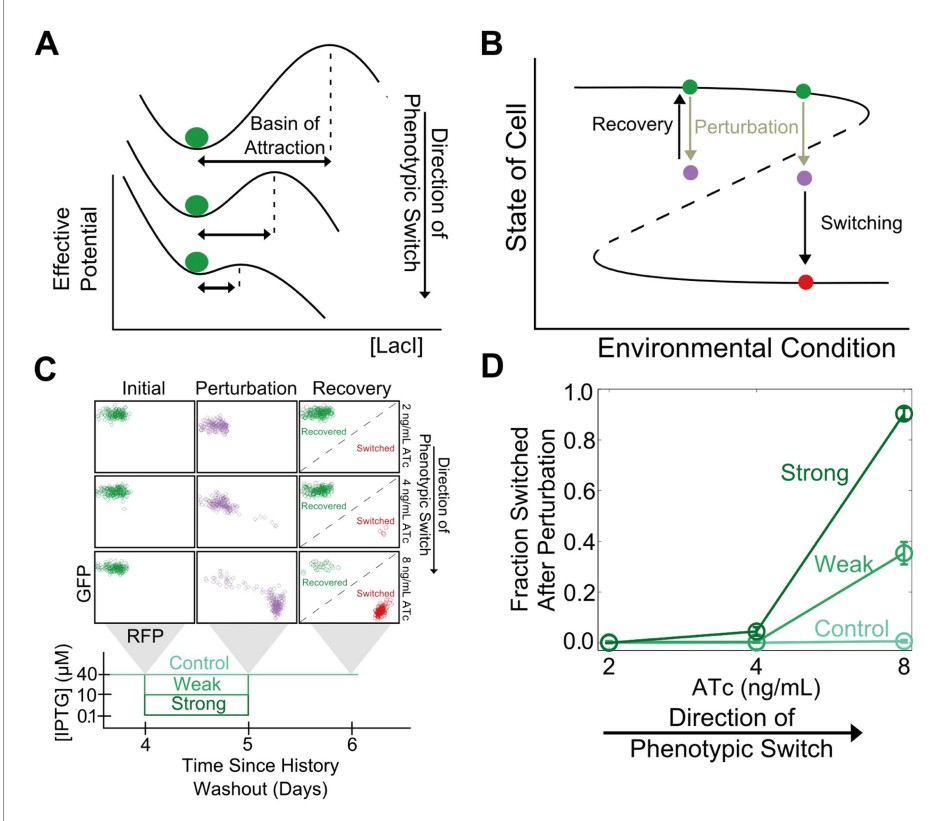

**Figure 2**. Cellular memory of the high GFP history in the toggle switch loses resilience to directional. (**A**) A schematic of how the effective potential changes and the basin of attraction shrinks approaching the critical transition. The size of the basin of attraction is determined by the distance between the stable and unstable fixed points. (**B**) Far from the critical transition, a perturbation temporarily depresses the value of GFP; the system recovers to its initial state after the perturbation is removed. Close to the critical transition, the same perturbation causes the system to cross the basin boundary into the alternative state. (**C**) 92 hr after history washout, cells at different distances from the phenotypic switch were exposed to a reduction in (IPTG) from 40 μM to 0.1 μM. Cells grew for 24 hr in this new condition. IPTG was then restored to 40 μM and cells were allowed to recover for 24 hr. Control cells were propagated with (IPTG) held fixed at 40 μM. (**D**) The fraction of cells that switched into a high RFP state in response to the perturbation is plotted as a function of distance from the tipping point. Two different strength perturbations, a weak (10 μM) and a strong (0.1 μM) are plotted. Error bars in **D** represent the standard error of three different samplings from forward scatter area (FSC-A) vs side scatter area (SSC-A) (see *Figure 2—figure supplement 2*).

The following figure supplements are available for figure 2:

**Figure supplement 1**. Cellular memory of the high RFP phenotypic state of the toggle switch loses resilience to directional perturbations.

**Figure supplement 2**. To control for cell size, tight gates on FSC-A vs SSC-A are selected for analysis.

To explore the directional perturbation we prepared cells in the GFP state by pre-growing them in IPTG. We then transferred the populations to multiple ATc concentrations for 4 days, leaving the cells in the high GFP state but at varying distances from the critical transition. At the end of the fourth day, the cells were then perturbed by decreasing the concentration of IPTG for 1 day before returning the IPTG concentration to its original value. Far from the critical transition (2 ng/ml ATc), the IPTG perturbation caused almost no cells to switch their phenotypic state (*Figure 2C*). However, close to the critical transition (8 ng/ml ATc), the perturbation caused a great majority of the cells to switch into a high RFP state. Even after the perturbation was removed and the initial conditions were restored, many cells remained in a high RFP state. As expected, the severity of the environmental perturbation

(i.e., the magnitude of reduction in IPTG) correlated with the fraction of cells that switched states in response to the perturbation (*Figure 2D*). Importantly, there was negligible switching (∼0.5%) into the high RFP state for control cells grown in constant IPTG concentrations, demonstrating that the phenotypic switching was indeed caused by the perturbation. Moreover, the lack of phenotypic switching in the absence of the perturbation also indicates that in all of these conditions the traditional measure of cellular memory—the lifetime of the state—would classify all of these conditions as being stable with a high degree of cellular memory. Similar results were observed in the other direction, as cells pre-grown in the RFP state approach the critical transition associated with sudden switching to the GFP state (*Figure 2—figure supplement 1*). Thus, phenotypic states in the toggle switch lose resilience to directional perturbations near a critical transition.

Cellular memory in development and cell cycle progression must be resilient against a wide range of different environmental perturbations. Given this, we wanted to explore whether memory in the toggle switch would lose resilience against generic perturbations approaching a critical transition. Cells from a high GFP history at different distances from the phenotypic switch (2, 4, and 8 ng/ml ATc) were perturbed in several different ways for 24 hr: heat stress, osmotic stress with sodium chloride, ethanol stress, and a glucose pulse. Remarkably, we observed a loss of resilience against all four of these generic perturbations (*Figure 3A,B*). Far from the critical transition (2 ng/ml ATc), there was little to no phenotypic switching in response to any of these 'generic' perturbations. However, close to the critical transition (8 ng/ml ATc) we observed nearly complete switching in all perturbations, despite the fact that there was essentially no switching (∼0.5%) in the absence of the perturbations. At a given distance from the phenotypic switch, increasing the strength of a generic perturbation increased the probability that cells would switch into the alternative state (*Figure 3—figure supplement 1*). The switching induced by the glucose perturbation can perhaps be understood by the fact that glucose shuts down expression of the entire system (LacI, TetR, GFP, and RFP) via catabolite repression of a GAL1 upstream activation sequence, thus pushing the cells toward a low GFP and low RFP state (*Gardner et al., 2000*; *Ellis et al., 2009*). The other three perturbations have much broader effects on the cell with no obvious connection to the toggle switch network being probed in our experiments. Cellular memory can therefore degrade near a critical transition for a wide range of different environmental perturbations (*Figure 3C*).

Complex dynamic systems near critical transitions are predicted to lose resilience to a specific class of perturbations: those that push the system toward the alternative stable state. However, the effect of a salt shock or a heat shock on the GFP output of the toggle switch is challenging to predict a priori. Notably, it is possible that some generic perturbations will stabilize the state of the cells and reduce their probability of switching. Indeed, we found that none of the four generic perturbations led to a loss of resilience in the transition from the high RFP state (*Figure 3—figure supplement 3*). This result is unsurprising given that the perturbations did not strongly push the genetic network toward the high GFP state (*Figure 3—figure supplement 3*). For the salt, ethanol, and heat stresses, the perturbation increased the RFP output of the cells, thereby stabilizing the state. Thus, phenotypic states in the toggle switch only loss resilience to perturbations that push the system toward the alternative state.

We reasoned that our experimentally-observed loss of resilience could be caused by a shrinking basin of attraction of the phenotypic state. To test this hypothesis, we used our perturbation experiments to estimate the location of the basin boundary in GFP-RFP space and see how it shifted for different concentrations of ATc (*Figure 4A*). For a given concentration of ATc and a chosen perturbation, we measured the fraction of cells that switched into the alternative state at the end of the recovery period. Analysis of the GFP-RFP distribution at the end of the perturbation and before the recovery period revealed that many cells were in an indeterminate state between the two stable phenotypic states. We ranked the cells according to the ratio of GFP expression to RFP expression and assumed that a cell would switch phenotypic states when its ratio of GFP expression to RFP expression fell below a critical threshold. This is equivalent to assuming that the basin boundary between the two phenotypic states takes the form of a line on a log–log plot. To justify this assumption, we used a simple mathematical model of the toggle switch (see *Figure 4—figure supplement 1* and 'Materials and methods'). Our modeling indicated that this assumption would hold as long as the promoter strengths of the two transcription factors in the toggle switch were approximately equal.

Using the assumption described above, we estimated the location of the basin boundary consistent with our experimentally observed fractions at the end of the recovery period. Encouragingly, we find

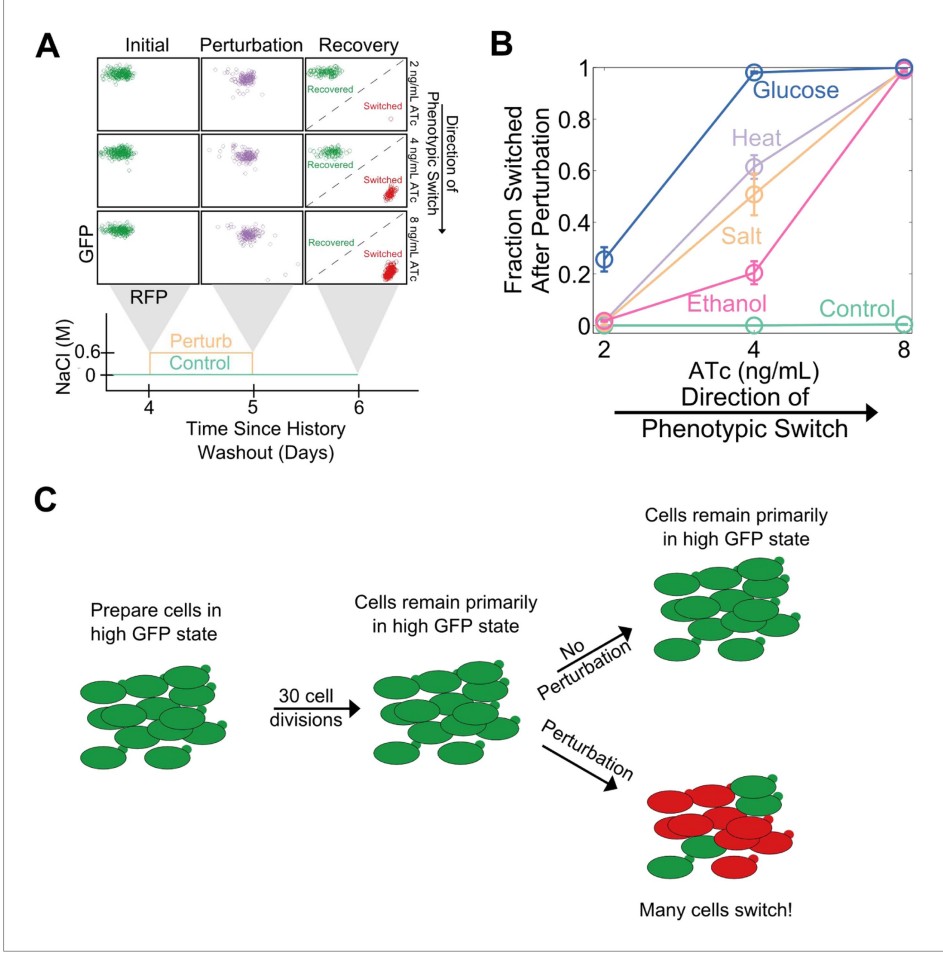

**Figure 3**. Cellular memory of the high GFP history in the toggle switch loses resilience to generic perturbations. (**A**) Cells were pre-grown in the high GFP state. 92 hr after history washout, cells at 2, 4, and 8 ng/ml ATc were exposed to an osmotic stress (600 mM NaCl). Cells grew for 24 hr in this new condition. The osmotic stress was then removed and cells were allowed to recover for 24 hr. Control cells were propagated with NaCl held constant throughout the whole time course. Growth media contains trace NaCl (2 mM). (**B**) The fraction of cells that switched into a high RFP state is plotted as a function of [ATc]. During the 24 hr perturbation period, cells were exposed to 6% ethanol (pink), 600 mM NaCl (peach), 37°C (violet), 0.2% glucose (blue), or no perturbation (teal). Error bars in **B** represent the standard error of three different samplings from FSC-A vs SSC-A. All results were replicated in a second independent experiment several weeks later (see *Figure 3—figure supplement 2*). (**C**) A schematic of the key findings from the perturbation experiments.

The following figure supplements are available for figure 3:

**Figure supplement 1**. Increasing the strength of a generic perturbation increases the probability that cells will switch into the alternative phenotypic state.

**Figure supplement 2**. A loss of resilience to generic perturbations was confirmed in a second independent experiment.

**Figure supplement 3**. Cellular memory of the RFP state of the toggle switch does not lose resilience to generic perturbations because salt, ethanol, and heat shocks stabilize the high RFP state.

that for a given concentration of ATc, the switching fraction for the eight different perturbations (weak and strong IPTG, weak and strong ethanol, weak and strong salt, heat, glucose) is well fit by a single separatrix (*Figure 4B*). Moreover, as ATc increases from 2 to 8 ng/ml, the location of the basin

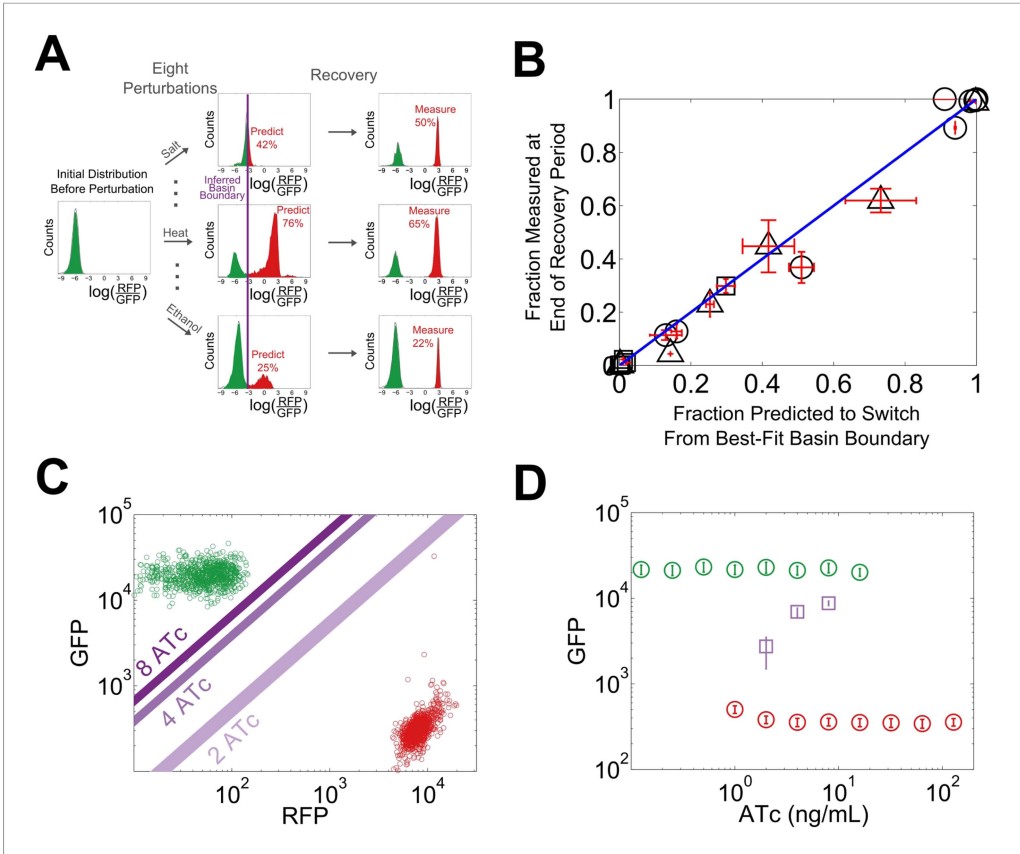

**Figure 4**. The loss of resilience is due to a shrinking basin of attraction of the phenotypic state. (**A**) By examining the RFP-GFP distribution at the end of the recovery period and comparing it to the end of the perturbation period, the basin boundary can be estimated. A cell is assumed to switch when its ratio of GFP to RFP expression falls below some threshold α, so the separatrix is a line with slope 1 and intercept α on a log–log plot. A simple model of the toggle switch supports this assumption (see *Figure 4—figure supplement 1*). For each [ATc], eight perturbations (10 μM IPTG, 0.1 μM IPTG, 37°C, 200 mM NaCl, 600 mM NaCl, 2% ethanol, 6% ethanol, and 0.2% glucose) were used to estimate α by minimizing the mean-squared deviation between the estimated and measured fractions. (**B**) The estimated fraction is compared to the measured fraction for [ATc] = 2 ng/ml (□), 4 ng/ml (△), and 8 ng/ml (○). (**C**) The unperturbed GFP-RFP distribution for cells at 0 ng/ml (green) and 128 ng/ml (red) is overlaid with the estimated separatrix from 2, 4, and 8 ng/ml. μ ± σ is shaded for each separatrix. (**D**) The location of the high GFP stable fixed point (green), unstable fixed point (purple), and low GFP stable fixed point (red) are plotted as a function of ATc. The system is bistable for intermediate ATc and monostable at low and high ATc. We assume that switching follows a line in log-space connecting the centroids of the two distributions in *Figure 4C* (see *Figure 4—figure supplement 2*). Error bars in all plots represent the standard error from three samplings from the FSC-A vs SSC-A distribution.

The following figure supplements are available for figure 4:

**Figure supplement 1**. A simple model of the toggle switch justifies the assumption that the basin boundary can be approximated as a line in LacI-TetR space.

**Figure supplement 2**. Cell switching paths approximately follow a line on a log–log plot connecting the stable fixed points.

**Figure supplement 3**. Increasing the duration of the perturbation increases the fraction of cells that switch phenotypes in a Gillespie simulation of the toggle switch.

**Figure supplement 4**. Not all perturbations induce switching in a Gillespie simulation of the toggle switch.

boundary gets closer to the location of the high GFP stable fixed point (*Figure 4C*). Knowing the location of the basin boundary allowed us to map the bifurcation (*Figure 4D*). Thus, the loss of resilience appears to be driven by a shrinking basin of attraction of the phenotypic state near the critical transition.

Given that the phenotypic states associated with our toggle switch lose resilience to perturbations near the critical transition, a natural question is whether, just as in other examples of critical transitions in complex systems such as ecosystem collapse, it is possible to develop warning indicators that this loss of resilience is taking place (*Dai et al., 2012*). For example, one might expect that the mean GFP of cells in the high GFP state would decrease with increasing ATc, thus potentially signaling that the critical transition is approaching (*Isaacs et al., 2003*). However, we find that the mean GFP of unswitched cells is approximately constant over the range of ATc concentrations in which we observe a loss of resilience (*Figure 1B* and *Figure 5A*). Researchers in a number of fields have explored early warning indicators based on a loss of stability near a critical transition (broadening of the effective potential as illustrated in *Figure 2A*) (*Scheffer et al., 2009*). This loss of stability would manifest in our experiments as an increase in the variation of GFP fluorescence among the population of (unswitched) cells. However, we find experimentally that there is no increase in variation within the population

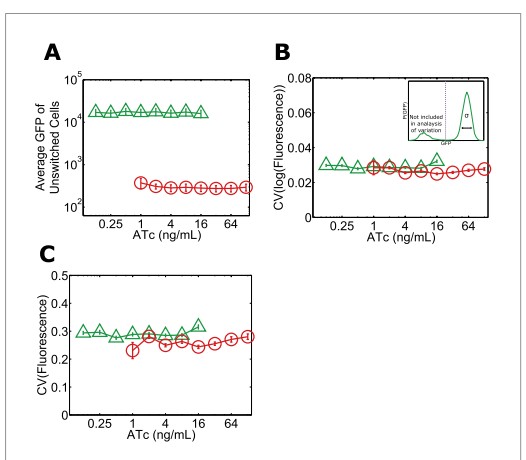

**Figure 5**. No significant change in mean fluorescence of the state and no significant increase in coefficient of variation approaching the critical transition. Yeast cells expressing the toggle switch were pre-grown in the high GFP state (green triangles) or the high RFP state (red circles) and then transferred to a range of ATc concentrations for 92 hr. (**A**) The mean GFP fluorescence of cells that have not switched from their pre-growth state is plotted against [ATc]. Above 16 ng/ml (for the GFP history) and below 1 ng/ml (for the RFP history), all of the cells have switched into the alternative state. To quantify population variability, the standard deviation normalized to the mean (coefficient of variation, i.e., 'CV') is plotted for the **B**, log-transformed and **C**, linear values of fluorescence. When calculating variation, only cells that remain in the state they were pre-grown in are analyzed (see inset in **B**). To minimize the effect of instrument noise in **B** and **C**, variation in RFP fluorescence is measured for the RFP history (similarly, variation in GFP is measured for the GFP history). Error bars in **A** represent the standard error of three samplings from FSC-A vs SSC-A. Error bars in **B** and **C** are standard errors from 200 bootstrap resamplings of the data.

approaching the critical transition, even very close to the transition where the GFP state is meta-stable (*Figure 5B,C*). The theoretically proposed early warning indicators based on local stability therefore fail to predict the critical transition in this gene network (*Menck et al., 2013*).

To explore the generality of our results, we chose to study the yeast GAL network, which displays an 'all-or-none' response in some sugar environments containing galactose (*Biggar and Crabtree, 2001*; *Song et al., 2010*). However, the wild-type GAL network exhibits only weak memory (*Song et al., 2010*). To expand the range of galactose concentrations for which the system has memory (*Acar et al., 2005*), we used a strain of yeast constitutively expressing the repressor GAL80 (which codes for a transcriptional repressor involved in one of the many feedback loops that stabilize memory in this network). To assess the state of the network, yellow fluorescent protein (YFP) expression was driven by a GAL1 promoter. Cells were pre-grown in high galactose (GAL ON) and then grown for a day in a range of galactose concentrations (*Figure 6A*). We then examined the resilience of the GAL network to perturbations at different distances from its critical transition, similar to our experiments with the toggle switch. For the GAL network, a glucose pulse served as a directional perturbation (due to catabolite repression [*Gancedo, 1998*]), and we again used heat, salt, and ethanol as generic perturbations. We found that the 'GAL ON' phenotypic state lost resilience to glucose and ethanol perturbations but not salt or heat (*Figure 6B*). Similarly, we performed experiments probing the resilience of the ON state in the *Escherichia coli* lac operon, where we again observed a loss of resilience to both directional perturbations and to the generic perturbation ethanol (*Figure 6—figure supplement 1*).

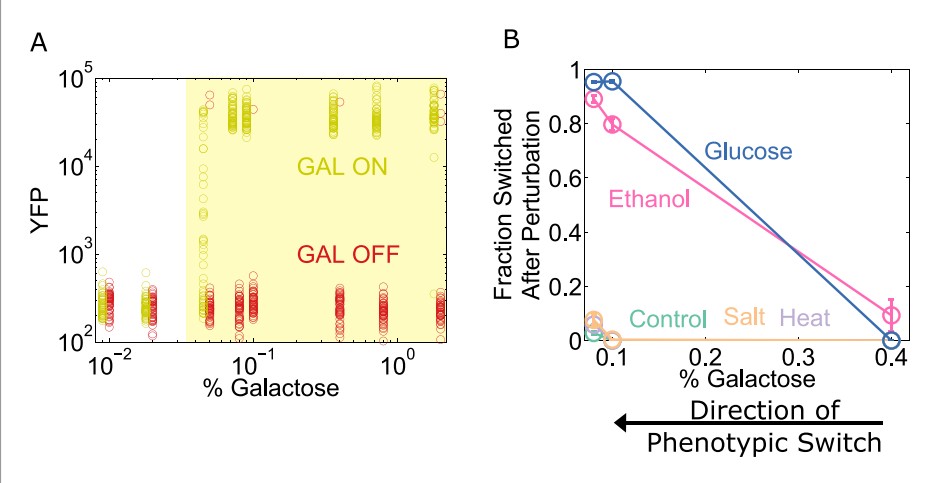

**Figure 6**. The yeast galactose network loses resilience to directional perturbations and the generic perturbation ethanol. (**A**) 25 hr after history washout, a gal80-inducible strain shows strong memory above 0.05% galactose. The two histories are offset for ease of viewing, and the region of memory is shaded in yellow. 50 cells at random are plotted from a tight gate on FSC-A vs SSC-A. (**B**) 25 hr after history washout, cells at 0.4%, 0.1%, and 0.08% galactose were exposed to 6% ethanol (pink), 600 mM NaCl (peach), 37°C (violet), 0.1% glucose (blue), or no perturbation (teal). Cells grew for 12 hr in this new condition. The perturbation was then removed and cells were allowed to recover for 12 hr. Control cells were propagated with fixed glucose, galactose, and temperature. The fraction of cells that switched into a low YFP state after the perturbation is plotted as a function of [galactose]. Error bars are standard errors obtained by bootstrap with 200 resamplings of the data.

The following figure supplement is available for figure 6:

**Figure supplement 1**. The *Escherichia coli* lactose network loses resilience to directional perturbations as well as to the generic perturbation ethanol.

These three genetic switches have widely different architectures, and operate in two different species. Our results thus indicate that a loss of resilience approaching a phenotypic switch could be a general property of multistable gene networks and that 'generic' or global perturbations, such as temperature or salt shocks, can cause widespread loss of cellular memory.

## Discussion

Here we have shown that brief perturbations in the extracellular environment can dramatically increase the rate of phenotypic switching from a highly stable memory state into an alternative state. We argue that this loss of resilience to perturbations near the bifurcation can be explained by a shrinking of the basin of attraction of the phenotypic state. By observing the GFP-RFP distribution at the end of the perturbation period and applying a simple threshold, we were able to accurately predict the fraction of cells that would return to the pre-perturbation memory state at the end of the recovery period. At a given distance from the bifurcation, the same basin boundary was able to accurately predict the switching fraction in response to eight different perturbations. The predictive success of this threshold is particularly impressive given that the different perturbations have dramatically different effects on the GFP-RFP distribution (see *Figure 4A*).

There is an interesting question of the precise mechanism by which the perturbations act on the system to push the cells to a new location in GFP-RFP space. Part of the answer is certainly that the perturbations cause a shift in the stability landscape underlying the phenotypic states. It is also possible that the perturbations cause an increase in noisy gene expression. Understanding what is occurring during the perturbation period, both biochemically and from a dynamical systems perspective, will be an area of investigation for future research.

The perturbations (salt stress, heat stress, etc) lasted for a significant duration: 24 hr, or roughly 3 to 8 cell divisions. It is interesting to ask whether we would have seen similar results if we had perturbed the

system for a shorter duration. To address this question, we performed Gillespie simulations using a very simple phenomenological model of the toggle switch (see *Source code 1*). Similar to our test tube experiments, we initialized the system in a high Lac state. The system was allowed to equilibrate for several generations, before we suddenly changed one of the parameters of the system (either the disassociation constant between the transcription factor and the DNA or the promoter strength). We performed this perturbation for a variable duration ranging from 0.2 generations to 20 generations. The system was then allowed to recover (by restoring the parameter to its initial value) for several generations, and we then determined what fraction of cells remained in a high Lac state. Two key observations emerged. (1) The degree of switching is set by both the intensity and duration of the perturbation. For dramatic perturbations, the minimum duration of perturbation to induce switching is approximately set by the cell division time (*Figure 4—figure supplement 3*). (2) Not all perturbations induced phenotypic switching. Of note, perturbations which stabilized the high Lac state induced negligible switching into the high Tet state (*Figure 4—figure supplement 4*). These observations held regardless of whether the perturbation was achieved by changing the disassociation constant or the promoter strength.

Understanding the stability of phenotypic states in gene networks remains an important challenge in biology. Here we have demonstrated that for robust cellular memory in natural contexts subject to environmental noise, low rates of stochastic switching from the phenotypic state is not sufficient. This is because the cellular memory must also be robust against environmental fluctuations and perturbations that are impossible to avoid. Here we found experimentally that several phenotypic states lost resilience to multiple environmental perturbations near a critical transition. Given that over time there will often be multiple kinds of environmental fluctuations and perturbations, our results argue that many forms of cellular memory will become destabilized near a critical transition. Our results provide a roadmap for exploring cellular memory and phenotypic switching in other contexts, from development to cancer progression.

## Materials and methods

### Strains

Toggle switch strains have been previously characterized (*Ellis et al., 2009*; *Wu et al., 2013*). Gal80-inducible strains have been previously described (*Acar et al., 2005*). LacY-YFP fusion strains have been previously described (*Choi et al., 2008*).

### Toggle switch experiments

Cells were grown in synthetic media (YNB and CSM—Trp—Leu; Sunrise Science, San Diego, CA, United States) containing 2% galactose and ATc/IPTG (Sigma Aldrich, St. Louis, MO, United States) as described in the text. Cells were grown in 3 ml cultures in 14 ml VWR culture tubes (VWR, Radnor, PA, United States) and diluted daily to prevent saturation. Cells were pre-grown for 24 hr in either 1 mM IPTG or 250 ng/ml ATc and then transferred to a range of ATc concentrations as described in the text. 20 μl of cells were harvested daily at OD 0.5, diluted in 180 μl PBS, and run immediately on a Miltenyi MACSQuant VYB flow cytometer (Miltenyi Biotec, San Diego, CA, United States). After 4 days of serial transfer, cells were transferred to the perturbation environment. Cells grew for 24 hr in this environment before being characterized via flow cytometry and diluted into the original environment they were in before the perturbation (hereafter, the 'recovery environment'). Cells grew for 24 hr in the recovery environment and then were characterized with flow cytometry. The basin boundary was estimated by assuming that cells switched phenotypes when the ratio of GFP to RFP expression fell below a critical threshold. All results were verified in two independent experiments carried out several weeks apart.

### GAL network experiments

Cells were pre-grown for 24 hr in either 0.5% galactose or 0.01% glucose (GAL ON and GAL OFF, respectively) plus 0.05 μg/ml doxycycline (Sigma Aldrich). Cells were then diluted into 0.01% glucose, 0.05 μg/ml doxycycline, and galactose concentrations ranging from 0 to 2%. Cells were diluted every 12 hr and the OD was kept below 0.01 to minimize consumption of the sugars. Cells were harvested every 12 hr, concentrated via centrifugation, and characterized immediately via flow cytometry. Cells were transferred to the perturbation environment 24 hr after history removal. They were then grown for 12 hr, characterized via flow cytometry and diluted into the recovery environment, grown for a further 12 hr, and characterized via flow cytometry.

## Lac network experiments

Cells were grown in M9 media (Sigma Aldrich) with 0.1% succinic acid as a carbon source. Cells were first pre-grown in either 0 or 100 μM TMG ('Lac OFF' and 'Lac ON,' respectively) for 20 hr. The history condition was then washed out, and the two populations were then separately transferred to a range of TMG concentrations. Cells grew for 21 hr. YFP expression was assayed using flow cytometry as a proxy for the state of the Lac network. A constitutively expressed RFP enabled for discrimination between cells and noise. 21 hr after history washout, cells were diluted into the perturbation environment and grown for 12 hr. After 12 hr of growth, YFP expression was again assayed using flow cytometry. At the same time, the cells were diluted into the recovery environment and grown for a further 12 hr. At the end of the recovery period, YFP expression was again assayed using flow cytometry.

## Data analysis

Flow cytometry data was analyzed utilizing the Gore lab's flow cytometry tool kit, which can be accessed at http://gorelab.bitbucket.org/flowcytometrytools/. The code for the Gillespie simulation of the toggle switch has been uploaded as a supplementary file.

## Theoretical model for the toggle switch

The rate of production of the two repressors is described by the following equations:

$$[\dot{LacI}] = \frac{P_{Tet}}{1 + \left(\frac{[TetR]}{K_{Tet}}\right)^2} - \gamma[LacI], \tag{1}$$

$$[\dot{TetR}] = \frac{P_{Lac}}{1 + \left(\frac{[LacI]}{K_{Lac}}\right)^2} - \gamma[TetR]. \tag{2}$$

The promoter strength is determined by the DNA sequence and has been previously characterized (*Ellis et al., 2009*). As a simple approximation, we treat ATc and IPTG as modulating $K_{Tet}$ and $K_{Lac}$. We assume that the proteins are stable, so the rate of transcription factor degradation is set by dilution via cell division.

Our primary goal for the model was to predict the shape of the basin boundary between the two phenotypic states, particularly in a system where the promoter strengths are asymmetric. *Figure 4—figure supplement 1* was generated by picking values of $K_{Tet}$ and $K_{Lac}$ so that the system exhibited bistability over a range of [LacI] and [TetR]. We initialized the system with many different [LacI] and [Tet] and allowed the system to evolve according to *Equations 1, 2* until the system reached a steady state.

For the equal promoter strengths, the following parameters were used:

$P_{Lac} = P_{Tet} = 50$
$K_{Lac} = K_{Tet} = 15$
$\gamma_{Lac} = \gamma_{Tet} = 0.5$

For the unequal promoter strengths, the following parameters were used:

$P_{Lac} = 30$
$P_{Tet} = 35$
$K_{Lac} = K_{Tet} = 15$
$\gamma_{Lac} = \gamma_{Tet} = 0.5$

For the Gillespie simulations to investigate the effect of perturbation duration on switching fraction, we again used:

$P_{Lac} = P_{Tet} = 50$
$K_{Lac} = K_{Tet} = 15$
$\gamma_{Lac} = \gamma_{Tet} = 0.5$

Similar to our test tube experiments, we initialized the system in a high Lac state. The system was allowed to equilibrate for several generations, before we suddenly changed one of the parameters of

the system (either the disassociation constant between the transcription factor and the DNA or the promoter strength). We performed this perturbation for a variable duration ranging from 0.2 generations to 20 generations. The system was then allowed to recover (by restoring the parameter to its initial value) for several generations, and we then determined what fraction of cells remained in a high Lac state.

## Acknowledgements

The authors thank James Collins, Xiao Wang, Tom Ellis, Xiaoliang Sunney Xie, and Alexander van Oudenaarden for strains. We thank Lei Dai, Eugene Yurtsev, David Healey, and the members of Gore Lab for insightful comments on the manuscript. KCA is supported by the Harvard Biophysics Program. This work was funded by an NIH New Innovator Award. JG is an Allen Distinguished Investigator, Pew Scholar in the Biomedical Sciences, Sloan Fellow, and NIH Director's New Innovator Awardee.

## Additional information

### Funding

| Funder | Grant reference | Author |
| --- | --- | --- |
| NIH Office of the Director (OD) | NIH New Innovator | Kevin Axelrod, Alvaro Sanchez, Jeff Gore |
| Harvard University | Biophysics Training Grant | Kevin Axelrod |

The funders had no role in study design, data collection and interpretation, or the decision to submit the work for publication.

### Author contributions

KA, Acquisition of data, Analysis and interpretation of data, Drafting or revising the article; AS, Conception and design, Acquisition of data, Analysis and interpretation of data, Drafting or revising the article; JG, Conception and design, Analysis and interpretation of data, Drafting or revising the article

## Additional files

### Supplementary file

• Source code 1. The code for the Gillespie simulation of the toggle switch has been uploaded as a supplementary file.

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
