## [Decision Letter]

Thank you for submitting your work entitled “Cellular Memory Becomes Increasingly Sensitive to Perturbations Approaching a Phenotypic Switch” for peer review at *eLife*. Your submission has been favorably evaluated by Naama Barkai (Senior editor) and three reviewers, one of whom, James Ferrell, is a member of the Board of Reviewing Editors.

The reviewers had quite an intensive discussion on the manuscript. The two main concerns were firstly the issue of novelty, in particular with respect to your previous 2011 paper, but also with respect to other synthetic biology papers that were published. Secondly, whether you in fact answered the question you pose regarding cellular memory. All the reviewers were concerned about these issues. The final decision was to ask you to revise the paper addressing the following points in particular.

1) Change the Title/Abstract and opening about “cellular memory” as the reviewers were all concerned that you never really address it afterward in the paper. It should be clear upfront in the Abstract that you explore a synthetic system and extend their previous findings.

2) Substantially enhance the explanation and discussion of the model.

3) Minor: re-examine your first-passage time data. The lack of an exponential first-passage time might indicate some meta-stable intermediate. You may want to discuss this.

The individual reviews are attached below, and please address their points as well. It is important that you realize that these reviews were written prior to the discussion between reviews, and that during the course of the discussion all reviews were in full agreement with the more critical comments of reviewer #2, who initially raised the two points I specified above. So please pay special attention to this review.

Reviewer #1:

This paper examines whether a synthetic double-negative feedback loop (LacI -| TetR -| LacI) becomes more sensitive to perturbations (IPTG, hyperosmolar stress) when it is operating close to its bifurcation point. The answer is yes. The increased switching in response to hyperosmolar stress is particularly neat.

This is not a huge step forward conceptually. But I cannot recall that anyone has previously demonstrated that the expected behavior really does occur. To me the bigger question is whether natural switches are subject to this sort of “hazard”, or whether the multiple interlinked feedback loops that always seem to be present serve to mitigate this behavior.

That said, I think this will serve as a nice example to which other synthetic and natural circuits can be compared. I support publication in *eLife* with only minor revisions.

The most important of my concerns has to do with the time scale of the perturbations. The authors use very long-lasting perturbations-24 h incubation in a new concentration of IPTG or in hyperosmolar stress. What happens when you use shorter perturbations? Does, for example, the perturbation need to be comparable in duration to the flipping time for the toggle switch?

*Reviewer #2*:

Axelrod et al. set out to analyze how cells store memories in the face of environmental fluctuations. The question posed is interesting and important and the Gore group is an outstanding research group. However, disappointingly, the study does not answer the question posed in an informative or mechanistic way.

The primary major concern is that the study is highly descriptive and lacks a clear mechanistic model that sheds light on the question of how cells store memories. For example, we would have liked to see identification of a fundamental mechanism or a new design principle on how memory storage is implemented. Instead, the study largely characterizes the history dependence of the toggle switch and the phenomenon of the toggle losing resilience. For example, in Figure 2, providing the precise model parameters that drive a shrinking basin of attraction might have helped to define a design principle. Figure 4 comes the closest to a mechanistic model but it really doesn't teach the reader very much about the underlying principles that drive the shrinking basin, instead it demonstrated consistency between model and experiment.

A second major concern is that the study uses a synthetic construct rather than a real biological system to examine a phenomenon very similar to what the Gore group has previously reported (Science, 2011). In my opinion, using synthetic constructs would not be a concern (even for a high-impact venue) if a new phenomenon was being presented and tested for feasibility. However, here the phenomenon being examined is very similar to what the Gore group has previously demonstrated in yeast (Science, 2011) so the present study comes across a little bit like a simulation, just using this synthetic toggle construct instead of a real system. On a related note, the authors' honesty about the lack of a CV change as an indicator of switching (Figure 5) was highly laudable.

We sympathize with the authors since in these types of studies it is often a long road to identifying the most relevant and impactful framing for presenting a set of findings. Nevertheless, with regret, we feel this manuscript needs substantial reworking in order to rise to a level where it is appropriate for publication in such a venue.

Reviewer #3:

This is another excellent study by the Gore laboratory. This is a first for me where the submitted manuscript is already in such good shape that any comments on my part would merely be nitpicking. Therefore, in the spirit of *eLife*, I suggest publication of this beautiful study, which provides evidence for how cellular memory can become destabilized by environmental variability near a critical transition.

1) Change the Title/Abstract and opening about “cellular memory” as it is not really addressed in the paper. Be clear upfront in the Title and the Abstract that this is a study of a synthetic circuit, not a natural circuit.

2) The study uses very long-lasting perturbations 24 h incubation in a new concentration of IPTG or in hyperosmolar stress. What happens when you use shorter perturbations? Does, for example, the perturbation need to be comparable in duration to the flipping time for the toggle switch? Some data on how long the perturbations need to be would be very helpful.

---

## [Author Response]

*1) Change the Title/Abstract and opening about* “*cellular memory*” *as the reviewers were all concerned that you never really address it afterward in the paper. It should be clear upfront in the Abstract that you explore a synthetic system and extend their previous findings*.

We agree that some the forms of cell memory we discuss in the Introduction may be viewed as an overgeneralization of the findings we present in the Results section. We agree that the majority of our results utilize a synthetic gene circuit, and we have updated the Title, Abstract, and Introduction (first paragraph) accordingly. We have reduced the scope of our introduction and removed references to topics such as pluripotent stem cells.

*2) Substantially enhance the explanation and discussion of the model*.

We introduced the model simply to justify our assumption that a cell switches phenotypes when the ratio of RFP to GFP expression falls below a critical threshold that depends on the value of [ATc]. Indeed, given the wide range of effects that the various perturbations exert on the cells’ transcription factor expression, we were somewhat surprised that we were able to obtain high-quality fits with such a simple model. To comply with the requests of the reviewers, we have expanded the explanation of the model (“We ranked the cells according to the ratio […] in the toggle switch were approximately equal”).

We also expanded the use of the model to investigate the minimum duration of perturbation required to induce phenotypic switching, as suggested by reviewer 1. A new paragraph has been added to the Discussion (“The perturbations (salt stress, heat stress, etc.) […] or the promoter strength”), and we also added a supplementary figure based on our simulations.

*3) Re-examine your first-passage time data. The lack of an exponential first-passage time might indicate some meta-stable intermediate. You may want to discuss this*.

We also found the first-passage time data interesting. We performed several experimental trials (10+ days) tracking the fraction of cells that remained in the history state as a function of time. The temporal trajectories varied somewhat from run-to-run but frequently displayed deviations from first-order kinetics. We emphasize that the OD was kept below .5 at all times and the cells never saturated. We have added a few sentences to the text to highlight the complexity of the switching kinetics and the possibility of metastable states (“…although interestingly we observe non-exponential switching kinetics, potentially indicating the presence of metastable states in the gene network (Figure 1—figure supplement 1 and Figure 1—figure supplement 2)”).

Reviewer #1:

*[…] The most important of my concerns has to do with the time scale of the perturbations. The authors use very long-lasting perturbations-24 h incubation in a new concentration of IPTG or in hyperosmolar stress. What happens when you use shorter perturbations? Does, for example*, *the perturbation need to be comparable in duration to the flipping time for the toggle switch?*

We also expanded the use of the model to investigate the minimum duration of perturbation required to induce phenotypic switching, as suggested by reviewer 1. The new paragraph has been added to the Discussion, and we also added a supplementary figure based on our simulations.

Reviewer #2:

*[…] The primary major concern is that the study is highly descriptive and lacks a clear mechanistic model that sheds light on the question of how cells store memories. For example, we would have liked to see identification of a fundamental mechanism or a new design principle on how memory storage is implemented. Instead, the study largely characterizes the history dependence of the toggle switch and the phenomenon of the toggle losing resilience. For example, in*
Figure 2*, providing the precise model parameters that drive a shrinking basin of attraction might have helped to define a design principle.*
Figure 4
*comes the closest to a mechanistic model but it really doesn't teach the reader very much about the underlying principles that drive the shrinking basin, instead it demonstrated consistency between model and experiment*.

We agree with the reviewer that finding new design principles for how memory is implemented in biological systems is a fascinating question and a major goal in the field. Our claim in this paper is that cells need to stay far from a critical transition in order for a gene network to be resilient to environmental perturbations. We would argue that this may qualify as a “design principle” that will be relevant for many different network architectures. The main takeaway from our work is that simply having a long lifetime (the traditional metric for memory in bistable gene networks) may be insufficient to ensure high-fidelity memory storage. This is because systems poised near a bifurcation could be highly sensitive to extracellular perturbations, which could abolish the memory of previous stimuli. Thus, high-fidelity memory in bistable gene circuits requires *both* a long lifetime *and* being far from a bifurcation. We believe that this conclusion is likely general and is relevant across a range of different specific implementations of cellular memory.

*A second major concern is that the study uses a synthetic construct rather than a real biological system to examine a phenomenon very similar to what the Gore group has previously reported (Science, 2011). In my opinion, using synthetic constructs would not be a concern (even for a high-impact venue) if a new phenomenon was being presented and tested for feasibility. However, here the phenomenon being examined is very similar to what the Gore group has previously demonstrated in yeast (Science, 2011) so the present study comes across a little bit like a simulation, just using this synthetic toggle construct instead of a real system. On a related note, the authors' honesty about the lack of a CV change as an indicator of switching (*Figure 5*) was highly laudable*.

First we would like to note that while the majority of our findings focus on the synthetic toggle switch system, we also replicated our findings in the GAL network in yeast and the lactose switch in *E. coli*. We therefore believe that our results likely have relevance for real biological gene networks.

Regarding the similarity of this study to our previous work, we note that our 2012 Science study focused on ecosystem collapse at the population level, which results from interactions between the ∼ 5 micron yeast cells. This study focuses on the dynamics of gene regulatory networks, which involve the interactions of proteins and DNA at the nanometer length scale, which is three orders of magnitude smaller than the phenomena that we previously studied. Moreover, the network of interactions in these two cases was very different. For these reasons, we do not believe that it is obvious which aspects of the phenomena will be the same or different between these systems. Indeed, we were quite surprised to find a lack of early warning indicators of the impending bifurcation (and we spent several months searching for them!). This highlights the point that complex dynamical systems have both universal *and* system-specific properties; quantitative experiments probing these different systems are therefore in our opinion essential.

Reviewer #3:

*1) Change the Title/Abstract and opening about* “*cellular memory*” *as it is not really addressed in the paper. Be clear upfront in the Title and the Abstract that this is a study of a synthetic circuit, not a natural circuit*.

We have made significant revisions to the Title, Abstract, and Introduction to address this request.

*2) The study uses very long-lasting perturbations 24 h incubation in a new concentration of IPTG or in hyperosmolar stress. What happens when you use shorter perturbations? Does, for example, the perturbation need to be comparable in duration to the flipping time for the toggle switch? Some data on how long the perturbations need to be would be very helpful*.

A similar question was raised by reviewer 1.